# Head-to-Head Comparison of FDG and Radiolabeled FAPI PET: A Systematic Review of the Literature

**DOI:** 10.3390/life13091821

**Published:** 2023-08-28

**Authors:** Priscilla Guglielmo, Pierpaolo Alongi, Lucia Baratto, Elisabetta Abenavoli, Ambra Buschiazzo, Greta Celesti, Miriam Conte, Rossella Filice, Joana Gorica, Lorenzo Jonghi-Lavarini, Helena Lanzafame, Riccardo Laudicella, Maria Librando, Flavia Linguanti, Francesco Mattana, Alberto Miceli, Laura Olivari, Leandra Piscopo, Cinzia Romagnolo, Giulia Santo, Antonio Vento, Fabio Volpe, Laura Evangelista

**Affiliations:** 1Veneto Institute of Oncology IOV-IRCCS, 35128 Padua, Italy; 2Nuclear Medicine Unit, A.R.N.A.S. Ospedali Civico, Di Cristina e Benfratelli, 90127 Palermo, Italy; alongi.pierpaolo@gmail.com; 3Department of Radiology, Division of Pediatric Radiology, Lucile Packard Children’s Hospital, Stanford University, Stanford, CA 94304, USA; lbaratto@stanford.edu; 4Nuclear Medicine Unit, Careggi University Hospital, Largo Brambilla 3, 50134 Florence, Italy; elisabettabenavoli@gmail.com; 5Nuclear Medicine Division, Santa Croce and Carle Hospital, 12100 Cuneo, Italy; ambra.buschiazzo@gmail.com; 6Nuclear Medicine Unit, Department of Biomedical and Dental Sciences and Morpho-Functional Imaging, University of Messina, 98122 Messina, Italy; celestigreta@gmail.com (G.C.); marialibrando@hotmail.it (M.L.); 7Department of Radiological Sciences, Oncology and Anatomo-Pathology, Sapienza University of Rome, 00185 Rome, Italy; miriam.conte@uniroma1.it (M.C.); goricajoana@gmail.com (J.G.); 8Unit of Nuclear Medicine, Biomedical Department of Internal and Specialist Medicine, University of Palermo, 90133 Palermo, Italy; filicer@libero.it (R.F.); riclaudi@hotmail.it (R.L.); 9Department of Nuclear Medicine, Fondazione IRCCS San Gerardo dei Tintori, 20900 Monza, Italy; lorenzo.jonghi@gmail.com; 10Department of Nuclear Medicine, West German Cancer Center, University Hospital Essen, 45147 Essen, Germany; helena.lanzafame@gmail.com; 11German Cancer Consortium (DKTK), Partner Site University Hospital Essen, 45147 Essen, Germany; 12Nuclear Medicine Unit, Department of Experimental and Clinical Biomedical Sciences “Mario Serio”, University of Florence, 50134 Florence, Italy; flavialinguanti@hotmail.it; 13Division of Nuclear Medicine, IEO European Institute of Oncology IRCSS, 20141 Milan, Italy; francesco.mattana@ieo.it; 14Nuclear Medicine Unit, Azienda Ospedaliera SS. Antonio e Biagio e Cesare Arrigo, 15121 Alessandria, Italy; albertomiceli23@gmail.com; 15Nuclear Medicine Unit, IRCCS Ospedale Sacro Cuore Don Calabria, 37024 Negrar, Italy; laura.oliv@hotmail.it; 16Department of Advanced Biomedical Sciences, University Federico II, 80138 Naples, Italy; lea-17-08@hotmail.it (L.P.); fabio.volpe@unina.it (F.V.); 17Department of Nuclear Medicine, “Ospedali Riuniti” Hospital, 60126 Ancona, Italy; cinzia.romagnolo@ospedaliriuniti.marche.it; 18Department of Experimental and Clinical Medicine, “Magna Graecia” University of Catanzaro, 88100 Catanzaro, Italy; giuliasanto92@gmail.com; 19Nuclear Medicine Department, ASP 1-P.O. San Giovanni di Dio, 92100 Agrigento, Italy; antvento@alice.it; 20Department of Biomedical Sciences, Humanitas University, 20090 Milan, Italy; laura.evangelista@hunimed.eu; 21IRCCS Humanitas Research Hospital, 20089 Milan, Italy

**Keywords:** fibroblast activation protein inhibitor (FAPI), FDG, PET/CT, PET/MRI, molecular imaging, hybrid imaging, new radiotracers, nuclear medicine, theranostics

## Abstract

FAPI-based radiopharmaceuticals are a novel class of tracers, mainly used for PET imaging, which have demonstrated several advantages over [^18^F]FDG, especially in the case of low-grade or well-differentiated tumors. We conducted this systematic review to evaluate all the studies where a head-to-head comparison had been performed to explore the potential utility of FAPI tracers in clinical practice. FAPI-based radiopharmaceuticals have shown promising results globally, in particular in detecting peritoneal carcinomatosis, but studies with wider populations are needed to better understand all the advantages of these new radiopharmaceuticals.

## 1. Introduction

The glucose analog fluorine-18 fluorodeoxyglucose, or [^18^F]FDG, is the most widely used radiopharmaceutical agent in positron emission tomography (PET) imaging for diagnosis, staging, prognosis, and treatment evaluation of many types of cancers [1]. The rationale for its use is based on the aerobic glycolysis (the so-called “Warburg effect”) of most cancer cells, which preferentially metabolize glucose by glycolysis, even in the presence of oxygen, leading to an increased accumulation of [^18^F]FDG in both primary tumor and distant metastases [2,3]. However, [^18^F]FDG shows well known limitations, such as high physiological uptake in normal tissues (e.g., brain), low uptake in some tumor types (e.g., well-differentiated neuroendocrine tumors, clear cell renal cell carcinoma, etc.), and lack of specificity as the cells involved in inflammatory response have high glycolytic activity [4,5]. In recent years the theranostic concept, involving the combination of diagnostic and therapeutic tools for the same molecular targets, has improved patient selection, led to better responses to treatment [6] and paved the way for the development of new “pan-tumor” agents, which could overcome the limitations of [^18^F]FDG; the growing knowledge of the tumor microenvironment (TME) [7,8], that is a heterogenous system composed of extracellular matrix (ECM) components, immune cells, fibroblasts, precursor cells, endothelial cells, and signaling molecules, that interact closely with tumor cells, contributing to the complex mechanism of tumorigenesis [9], further contributed to the development of new probes. Tumor cells gradually create mechanisms to escape immune surveillance, also known as “cancer immunoediting” [10,11,12]; during this dynamic process, nearby macrophages and fibroblasts are converted into tumor-associated macrophages (TAMs) and cancer-associated fibroblasts (CAFs). Thus, CAFs promote the growth and progression of tumors by boosting tumor cell proliferation, migration, invasion, and angiogenesis through immunosuppressive action and the generation of mediators [13,14,15]. CAFs are characterized by a high expression of fibroblast activation protein (FAP), a type II transmembrane serine protease belonging to the dipeptidyl peptidase 4 family, which is associated with ECM regulation [16]. FAP is highly overexpressed on the membrane of CAFs in approximately 90% of epithelial-derived tumors, as in the case of tissue damage, remodeling, or chronic inflammation, and, therefore, in benign conditions [17,18,19,20]. In contrast to CAFs, FAP expression is low or absent in normal tissues [21]. Thus, considering the high expression of FAP on the cell surfaces of activated CAFs and its limited expression in normal tissue, FAP-targeting ligands based on FAP inhibitors (FAPI) have recently been introduced [22,23], radiolabelled with either ^68^Ga or ^18^F, and several further FAPI variants were designed to increase tumor uptake and retention of these tracers in malignant cells [24]. The most widely used [^68^Ga]-labeled FAPI are characterized by rapid and stable tracer accumulation in target lesions [25]. On the other side, labeling FAPI with ^18^F, that has a longer half-life and lower positron energy than ^68^Ga (half-life of 110′ and positron energy of 0.65 MeV vs. 68′ and 1.9 MeV), would reduce the costs of production and facilitate its distribution together with improved spatial resolution [26]. Furthermore, besides the use of radiolabelled FAPI tracers for diagnostic purposes, DOTA chelator in the molecular structure allows the coupling of FAPI molecules with either alpha- or beta-emitting radioisotopes (i.e., Lutetium-177, Yttrium-90, Actinium-225), exploiting their potential role as a new anti-cancer radioligand therapy (RLT) in a theranostic perspective [23,24,27]. Like [^18^F]FDG, wound healing, fibrosis and inflammation can be visualized using FAPI-based tracers PET imaging [25]. The purpose of this systematic review is to perform a head-to-head comparison between the performances of [^18^F]FDG and FAPI-based tracers PET/computed tomography (PET/CT) or PET/magnetic resonance imaging (PET/MRI) in various tumors to estimate the potential value of this new class of radiopharmaceuticals as a valid alternative to [^18^F]FDG, highlighting their strengths and limitations.

## 2. Materials and Methods

A systematic literature search up to May 2023 was conducted in the PubMed and Scopus databases to find relevant articles comparing the diagnostic value of FAPI vs. [^18^F]FDG PET in oncology using the following keywords: “FDG”, “FAPI”, “PET” and “FAPI AND FDG comparison” or ‘’head-to-head FAPI AND FDG PET” or “FAPI AND FDG”. The inclusion criteria for papers’ selection were original articles, clinical studies, use of English language only and dual tracers PET imaging. Moreover, we excluded articles with mixed tumors from different anatomical sites.

The literature search was conducted by four investigators (E.A., R.F., L.O., H.L.), who independently reviewed titles and abstracts of the retrieved articles, ruling out the studies that did not fulfill the above-mentioned inclusion criteria, together with review articles, editorials, letters, comments, or case reports that were also excluded from the collection. Disagreements were solved with a consensus. Subsequently, the full text of each included study was reviewed, collecting relevant information (authors’ names, journal, year of publication, country of origin, PET tracer, number of participants, clinical settings, and study design). The methodological quality of the studies was critically assessed by two investigators (P.A. and A.V.) by using the Critical Appraisal Study Programme CASP for diagnostic performances. 

## 3. Results

Figure 1 (PRISMA flow-chart) shows the flowchart of the study selection process. After removing duplicates, a total of 67 papers were available to be reviewed. Fifteen studies were excluded from the analysis (three reviews, two case reports, one pre-clinical trial, and nine mixed-tumor articles). Fifty-two studies were deemed eligible. To summarize the current evidence, the studies were subdivided based on their content into three main categories following an increasing order in terms of [^18^F]FDG avidity as follows: (1) the utility of FAPI in tumors with low [^18^F]FDG uptake; (2) the role of FAPI in case of variable [^18^F]FDG uptake; and (3) the utility of FAPI in tumors with high [^18^F]FDG accumulation. A schematic representation of the studies and their advantage in terms of detection rate and/or accuracy is reported in Figure 2; Table 1 summarizes the main characteristics of the included studies.

Of these 52 articles, 41 papers were focused on [^68^Ga]Ga-DOTA-FAPI-04 (Figure 3A), two papers on [^18^F]FAPI-42 (Figure 3B), five papers on [^68^Ga]Ga-FAPI-46 (Figure 3C), one on [^68^Ga]Ga-DOTA.SA.FAPI (Figure 3D) and in three studies, only the radioisotope used (^68^Ga or ^18^F) was declared.

The risk of bias for the 52 studies included in the meta-analysis was scored as low by using QUADAS-2. No publication bias was detected (Figure 4).

### 3.1. Tumors with Low [^18^F]FDG Uptake

#### 3.1.1. Thyroid Cancer

Thyroid cancer (TC) is the most prevalent endocrine malignant neoplasm [80], mainly represented by differentiated TCs (DTCs). The standard of care for DTC is surgery followed by radioactive iodine (RAI, i.e., iodine-131) treatment in case of intermediate- to high-risk disease [81]. However, 5–15% of DTC and 50% of metastatic DTC develop a radioactive iodine-refractory condition, which loses avidity to ^131^I and is prone to aggressiveness [82]. The American Thyroid Association guidelines recommend the use of [^18^F]FDG PET/CT for detecting tumor recurrence and metastases in radioactive iodine-refractory DTC [83]. However, the sensitivity of [^18^F]FDG PET/CT for detecting recurrence and metastasis within the RAI-refractory DTC varies from 68.8 to 82%. Furthermore, a false-negative rate of 8–21.1% has been reported in patients with thyroglobulin (Tg) elevation and negative iodine scintigraphy (TENIS), which further complicates the treatment of metastatic DTC [83,84]. The efficacy of [^68^Ga]Ga-FAPI PET/CT in detecting lesions and guiding radioligand therapy of TC remains controversial. Some studies have suggested that low uptake values of [^68^Ga]Ga-FAPI or [^68^Ga]Ga-FAPI-negative lesions are observed in TC [16], whereas other studies have indicated that [^68^Ga]Ga-FAPI PET/CT is a promising tool for detecting metastatic TC [85,86,87,88]. In a head-to-head comparison between [^18^F]FDG and [^18^F]FAPI-42 PET/CT, Mu and colleagues [28] evaluated eleven patients with DTC and increased Tg or anti-Tg antibodies and found that, considering a total of ninety positive lesions detected in seven patients using both modalities, all positive lesions showed statistically higher uptake of [^18^F]FDG than [^18^F]FAPI-42, but the maximum standardized uptake value (SUVmax) of [^18^F]FAPI-42 was higher than that of [^18^F]FDG in local recurrences and lymphatic lesions [28]. Similarly, Fu et al. [29], in 35 DTC, found that the SUVmax of the lesions detected by [^68^Ga]FAPI-04 was higher than the SUVmax derived from [^18^F]FDG images in the metastatic lateral compartment, axillary, mediastinal lymph nodes, and pulmonary metastases. Furthermore, the same study proved that [^68^Ga]FAPI-04 PET/CT had a higher sensitivity than [^18^F]FDG PET/CT for depicting neck lesions (83% vs. 65%, *p* = 0.01) and distant metastases (79% vs. 59%, *p* = 0.001) [29].

#### 3.1.2. Liver and Biliary Tract Cancer

Liver cancer is the second leading cause of cancer-related death and represents a growing burden worldwide. Hepatocellular carcinoma (HCC), accounting for 70–80% of all liver cancer cases, is the most common pathological type of primary liver cancer [89]. Intrahepatic cholangiocarcinoma (ICC), the second-most common primary liver malignancy, comprises 10% of all primary liver cancers [90]. The use of [^18^F]FDG PET/CT in liver malignancies is controversial as studies indicate that up to 40–50% of HCCs are hypo- or isometabolic ([^18^F]FDG non-avidity), especially in case of low-grade HCC, which masks the malignant lesion and raises the false-negative rate [91]. Differently, as mentioned above, FAP is overexpressed in the CAFs of 90% of epithelial carcinomas, including primary and metastatic liver cancer [92]. 

A recent study by Rajaraman et al. [30] evaluated the performance of [^68^Ga]Ga-FAPI-04 PET/CT in a prospective cohort of 41 patients suspected to have HCC or cholangiocarcinoma (CC), subsequently found positive for malignancy (*n* = 31, of which 15 metastatic) or negative (*n* = 10) and compared with that of [^18^F]FDG PET/CT studies. For overall diagnosis of primary disease, [^68^Ga]Ga-FAPI-04 PET/CT resulted superior to [^18^F]FDG PET/CT with a sensitivity, specificity, and accuracy of 96.8% vs. 51.6%, 90% vs. 100%, and 95.1% vs. 63.4%, respectively. In particular, this study highlighted that [^68^Ga]Ga-FAPI-04 can be useful in the case of mucinous adenocarcinoma, which is poorly FDG-avid [31].

Another prospective study by Zhang et al. [32] aimed to investigate the potential utility of [^18^F]FAPI PET/CT for the assessment of non-FDG-avid focal liver lesions (FLLs) in a cohort of 37 patients (25 patients with liver malignancies and 12 with benign lesions). Indeed, in patients with non-FDG-avid liver malignancies (including HCC, ICC, and liver metastatic FLL), [^18^F]FAPI PET was able to differentiate benign and malignant disease with high diagnostic accuracy (83.8%). [^18^F]FAPI PET detected the majority of malignant FLLs in this lesion-based analysis, resulting in a high detection rate of 98.1%; moreover, the SUVmax and lesion-to-background ratio (LBR) of [^18^F]FAPI PET were considerably higher in malignant than in benign FLLs. Moreover, [^18^F]FAPI PET showed a high detection rate of 97% for HCC and 100% for other liver malignancies [32]. 

The aim of the retrospective analysis by Wang et al. [33] was to compare the diagnostic performances of [^68^Ga]Ga-FAPI-04 and [^18^F]FDG PET/CT in HCC and to assess factors associated with [^68^Ga]Ga-FAPI-04 uptake. A total of 35 intrahepatic lesions in 25 patients with HCC were analyzed. [^68^Ga]Ga-FAPI-04 PET/CT showed a higher sensitivity than [^18^F]FDG PET/CT in detecting intrahepatic HCC lesions (85.7% vs. 57.1%, particularly in patients with cirrhosis, low alpha-fetoprotein, multiple HCCs, and non-serious microvascular invasion), small (≤2 cm in diameter; 68.8% vs. 18.8%) and well- or moderately-differentiated HCCs (83.3% vs. 33.3%). The SUVmax was comparable between [^68^Ga]Ga-FAPI-04 and [^18^F]FDG, but the LBR was significantly higher for the [^68^Ga]Ga-FAPI-04 group compared with the [^18^F]FDG one. The SUVmax and LBR in [^68^Ga]Ga-FAPI-04 positive lesions were only associated with tumor size. Lymph node metastasis in one patient with poorly differentiated HCC showed a strong uptake of [^68^Ga]Ga-FAPI-04 but an undetectable uptake of [^18^F]FDG; furthermore, [^68^Ga]Ga-FAPI-04 PET/CT detected a small metastatic lesion that was not revealed by [^18^F]FDG PET/CT in another HCC patient with extensive peritoneal dissemination [33].

In a study by Sahin et al. [93], the two tracers were compared for the detection of liver metastases in 31 patients (fifteen colorectal, nine pancreas, four stomach and three other cancers). Patient-based sensitivities of [^68^Ga]Ga-DOTA-FAPI PET/CT and [^18^F]FDG PET/CT were 96.6% and 70.8%, whereas lesion-based sensitivities were 96.8% and 80.2%, respectively. The detection rate was significantly higher on [^68^Ga]Ga-DOTA-FAPI PET/CT compared to [^18^F]FDG PET/CT. No statistically significant difference was found in terms of SUVmax, while there was a significant difference in favor of [^68^Ga]Ga-DOTA-FAPI for all tumor subgroups in terms of metastatic lesion/liver uptake ratios [93].

A retrospective study by Siripongsatian et al. [34] compared tumor detection rates of [^68^Ga]Ga-FAPI-46 PET/CT with [^18^F]FDG PET/MRI in 27 patients with liver malignancies (13 CC, 14 HCC). All intrahepatic lesions detectable on MRI were also detected on [^68^Ga]Ga-FAPI-46 PET/CT (sensitivity of 100%), whereas the sensitivity of [^18^F]FDG PET was only 58%. [^68^Ga]Ga-FAPI-46 PET/CT revealed more positive lymph nodes and visceral metastases than [^18^F]FDG PET, especially in the brain and in the intra-abdominal (regional) and peritoneal lymph nodes. 

A work by Guo et al. [35] aimed to evaluate the potential utility of [^68^Ga]Ga-FAPI-04 PET/CT for diagnosing primary and metastatic lesions in patients with liver cancer, also in comparison with contrast-enhanced CT (ceCT), liver MRI, and [^18^F]FDG PET/CT in 34 patients diagnosed with/or suspected hepatic lesions. Out of thirty-four patients, twenty, twelve, and two subjects presented with HCC, ICC, and benign hepatic nodules, respectively. The sensitivity of ceCT, MRI, [^68^Ga]Ga-FAPI-04, and [^18^F]FDG/CT for detecting primary liver tumors were 96%, 100%, 96%, and 65%, respectively. Regarding the diagnosis of all intrahepatic lesions, the per-lesion detection rate of [^68^Ga]Ga-FAPI-04 PET/CT was slightly lower than that of MRI (85% vs. 100%, *p* = 0.34) and significantly higher than that of [^18^F]FDG PET/CT (85% vs. 52%, *p* < 0.001). Regarding the diagnosis of all malignant lesions (including extrahepatic disease), the tumor detection rate of [^68^Ga]Ga-FAPI-04 PET/CT was 87.4%, resulting significantly higher than that of [^18^F]FDG PET/CT (65%, *p* < 0.001). 

A prospective pilot study by Shi et al. [36] included sixteen HCC in fourteen patients and four ICC in three patients, determined by histology (*n* = 14) and clinical examinations (*n* = 3). Based on a visual analysis, seventeen patients presented elevated [^68^Ga]Ga-FAPI-04 uptake (sensitivity and specificity 100%), while only seven patients presented [^18^F]FDG avid tumors (sensitivity 58.8%, specificity 100%). [^68^Ga]Ga-FAPI-04 PET/CT identified 17 extrahepatic metastases vs. 13 in [^18^F]FDG PET/CT. The SUVmax and LBR max (of both HCC and ICC) in [^68^Ga]Ga-FAPI-04 PET/CT were significantly higher than [^18^F]FDG. Moreover, ICC patients showed higher levels of [^68^Ga]Ga-FAPI-04 uptake in the primary hepatic lesions compared to the extrahepatic metastases [36]. 

A recent analysis by Pabst et al. [31] evaluated the accuracy of [^68^Ga]Ga-FAPI-46 PET/CT for CC staging and management guidance in 10 patients (six with ICC and four with extrahepatic CC; six with G2 tumor and four with G3 tumor). All subjects underwent [^68^Ga]Ga-FAPI-46 PET/CT and conventional CT, and nine out of ten patients were sent to an additional [^18^F]FDG PET/CT. Overall, 22 lesions were detected across all modalities, including primary tumors, lymph node metastases, and distant metastases. All primary tumors were detected by all three imaging modalities. [^68^Ga]Ga-FAPI-46 PET/CT reached the highest detection rate for lymph nodes and distant metastases; the SUVmax was significantly higher for [^68^Ga]Ga-FAPI-46 than for [^18^F]FDG PET/CT for primary lesions and distant metastases; no significant difference was noted for lymph node metastases. Furthermore, tumor uptake for both tracers was analyzed in relation to tumor grade and location (intrahepatic and extrahepatic), showing that G3 tumors had significantly higher [^68^Ga]Ga-FAPI-46 uptake than G2, whereas the SUVmax was not significantly different between intra- and extra-hepatic CC. In addition, in one patient with an unknown primary, [^68^Ga]Ga-FAPI-46 PET/CT localized the tumor, and the subsequent biopsy with immunohistochemical (IHC) analysis led to a CC diagnosis [31]. 

Finally, a single-center prospective clinical study from Lan et al. [37] evaluated 18 participants with primary or recurrent biliary tract cancers. The sensitivity of [^68^Ga]FAPI PET/CT was higher than that of [^18^F]FDG PET/CT for detecting primary tumors (100% vs. 81%), nodal (98% vs. 83%), and distant metastases (100% vs. 79%). [^68^Ga]FAPI PET/CT resulted in new oncologic findings in ten out of eighteen participants and upgraded tumor staging or restaging in five out of eighteen participants compared with [^18^F]FDG PET/CT. Furthermore, [^68^Ga]FAPI PET/CT demonstrated higher sensitivity than [18F]FDG PET/CT in inflammatory processes secondary to tumor-related obstruction (88% vs. 13%) [37].

#### 3.1.3. Peritoneal Carcinomatosis

The presence of peritoneal neoplastic spread, also defined as peritoneal carcinomatosis (PC), impacts tumor staging, and it is one of the most significant prognostic indicators in several malignancies, especially in ovarian cancer and colorectal carcinoma, in which the prognostic significance of peritoneal spread is more important than tumor extent or lymph node involvement [94]. Unfortunately, the depiction of PC is considered a challenge due to its complex anatomical configuration and the extensive surface area that may host typically small or nodular tumor deposits [94].

It has been reported that the accuracy of [^18^F]FDG PET/CT in identifying PC is generally poor due to the physiological [^18^F]FDG uptake observed in the intestinal tract, the low [^18^F]FDG affinity shown by some types of gastrointestinal tract carcinomas (such as mucinous carcinoma and signet ring cell carcinoma) and the small size/volume of the lesions [38], due to the partial-volume effect which hampers their detection in a context of high diffuse uptake. On the other hand, post-surgical phenomena or post-radiation changes can lead to higher [^18^F]FDG uptake, eventually masking lesions’ uptake [38].

Differently, it has been observed that [^68^Ga]Ga-DOTA-FAPI-04 images are usually characterized by very low rates of non-specific uptake in the peritoneal cavity and a high LBR, which resulted in a superior sensitivity compared to [^18^F]FDG PET/CT in both quantitative and visual evaluations of PC (97–100% vs. 53–71%, respectively) [39]. Similar results were obtained by Elboga et al. [40] and Zhao et al. [38], who proved that [^68^Ga]Ga-DOTA-FAPI-04 PET/CT was superior to [^18^F]FDG PET/CT in terms of detection of peritoneal involvement with high image quality, as well as that of the primary tumor and other metastatic foci. Additional findings about the advantages of using FAPI over [^18^F]FDG are reported in the following paragraphs for cancers that can determine a peritoneal widespread.

#### 3.1.4. Soft Tissue Sarcoma

In the management of soft tissue sarcomas (STS), [^18^F]FDG PET/CT is not generally recommended due to a lack of sensitivity among some subtypes of sarcomas, particularly low-grade sarcomas [95,96,97]. As FAP is overexpressed in CAFs in the stroma of many subtypes of STS (e.g., fibrosarcoma, malignant fibrous histiocytoma, and liposarcoma) [98], in the study from Gu et al. [41] the authors investigated the role of [^68^Ga]Ga-DOTA-FAPI-04 in 45 patients affected by recurrent STS. Compared to [^18^F]FDG, [^68^Ga]Ga-DOTA-FAPI-04 PET/CT detected more lesions (275 vs. 186) and outperformed in sensitivity, specificity, positive and negative predictive values, and accuracy for the diagnosis of recurrent lesions [41]. Thus, [^68^Ga]Ga-DOTA-FAPI-04 demonstrated to be a new promising imaging modality for recurrent surveillance of STS regardless of tumor grade. 

### 3.2. Tumors with Variable [^18^F]FDG Uptake

#### 3.2.1. Breast Cancer

Breast cancer (BC), which is the prevalent form of cancer among women, represents nearly a quarter of all cancer cases affecting women globally (The Centers for Disease Control and Prevention. Available online: https://www.cdc.gov/cancer/breast/basic_info/index.htm) (accessed on 8 June 2023). Among the imaging modalities available for BC evaluation, [^18^F]FDG PET/CT is widely used for diagnosis, staging, response assessment, and prognostication, even though in case of tumors with specific biological features (i.e., lobular carcinoma or low-grade breast tumors) it may show a limited sensitivity [42,99]. As compared to other conventional imaging modalities, the sensitivity of [^18^F]FDG is remarkable, up to 97%; however, it exhibits a rather low specificity of 77% due to the high false-positive results in inflammatory lymph nodes and specifically low detection rates in micrometastases and sclerotic healed bone lesions [100]. Huang et al. [43] observed FAP expression in BC but not in normal breast tissue and proved that significant FAP expression was paralleled by increased tumor growth rates in a mouse model of human BC. Subsequently, other studies found [^68^Ga]Ga-FAPI-04 PET/CT useful in the detection of BC, lymph node, hepatic, bone, and cerebral metastases [43,44].

Some comparative studies [42,43,44,45] demonstrated that [^68^Ga]FAPI PET/CT was superior to [^18^F]FDG PET/CT in finding more lesions, showing higher tumoral activity retention and, consequently, high LBR. In the treatment-naive patient scenario examined by Zheng et al. [44], the accuracy for the diagnosis of negative axilla status of [^68^Ga]Ga-FAPI-04 and [^18^F]FDG PET/CT was 85.7% vs. 42.9%, respectively. FAPI showed higher accuracy in detecting primary tumors and lymph nodes, but no significant difference was shown between the two tracers in detecting distant metastasis. However, Kömek et al. [43] showed higher SUVmax values in lung and bone metastases at [^68^Ga]Ga-FAPI-04 PET/CT compared with [^18^F]FDG, but no significant difference in the SUVmax in hepatic metastases from BC (*p* < 0.005). In this setting, a limitation of FAPI PET emerged from the study of Elboga et al. [45], where [^68^Ga]Ga-FAPI-04 PET/CT showed more false-positive findings compared with [^18^F]FDG PET/CT due to various fibrotic processes such as granulomatous disease, myelofibrosis, liver cirrhosis and in lymph nodes harboring tuberculosis. 

Finally, in the study from Ballal et al. [42] [^68^Ga]Ga-DOTA.SA.FAPI scan identified more abnormal lesions than [^18^F]FDG in all the primary and metastatic sites with a maximum marked difference in the primary site (88.6% vs. 81.8%; *p* = 0.001), lymph nodes (89.1% vs. 83.8%; *p* = 0.0001), pleural metastases (93.3% vs. 73%; *p* = 0.096) and brain metastasis (100% vs. 59.5%; *p* = 0.0001). Nevertheless, the patient-based analysis showed no significant difference between the two modalities.

#### 3.2.2. Ovarian Cancer

Ovarian malignancies are the leading cause of cancer-related deaths among female reproductive organ cancers; these malignancies are often detected at an advanced stage due to non-specific symptoms, leading to poor outcomes for patients [46]. Accurate imaging evaluation, both at staging and in the later course of the disease, is, therefore, of crucial importance to tailor the best treatment strategy for the patients. In these settings, however, [^18^F]FDG PET/CT exhibits low to moderate sensitivity in detecting lesions in the diaphragm, liver capsule, and intestine due to physiological uptake and the relatively small size of these lesions [47]. 

Currently, two studies compared [^68^Ga]Ga-FAPI-04 and [^18^F]FDG PET/CT in the detection of ovarian malignancies. In 27 patients with suspected or previously treated ovarian cancer, Zheng et al. [46] compared both tracers’ uptake in primary tumors, lymph nodes and metastases using the SUVmax and LBR. [^68^Ga]Ga-FAPI-04 was able to detect 100% of the primary lesions and lymph nodes, while [^18^F]FDG only 78% and 80%, respectively. However, there was no significant difference in terms of SUVmax among both tracers, while the LBR was higher in [^68^Ga]Ga-FAPI-04, especially in peritoneal and pleural metastases [46].

Liu et al. [47] compared the two radiopharmaceuticals in 29 patients with platinum-sensitive recurrent ovarian cancer. Inconsistency between the two scans was seen in 75.86% of patients, and treatment strategies were, therefore, changed in 5 patients. Four patients who showed biochemical recurrence had detectable lesions at [^68^Ga]Ga-FAPI-04 scans. Notably, the smallest diameter of [^68^Ga]Ga-FAPI-04 uptake lesions was 0.3 cm. In conclusion, [^68^Ga]Ga-FAPI-04 PET/CT can have an impact on treatment strategies and help identify suitable candidates for surgery. However, at [^68^Ga]Ga-FAPI-04 PET/CT, false-positive results can occur. In fact, in the above-mentioned study, two lesions detected after surgery were confirmed to be postoperative changes characterized by fibrous tissue hyperplasia and calcium salt deposition, localized in the omentum (located posterior to the surgical incision) and in the abdominal wall (at the surgical incision site) [47]. 

#### 3.2.3. Gastric Cancer

Gastric cancer is the fifth most common cancer, with 1,089,103 new cases in 2020, and it represents the fourth cause of death in the same year [80]. The importance of an early and meticulous diagnosis is vital for the prognosis of patients affected by gastric adenocarcinoma (GAc). However, there is no clear consensus about the role of [^18^F]FDG in gastric cancer staging due to its low specificity and sensitivity related to the physiological uptake in gastric walls and inflammatory conditions and low avidity in diffuse-type, mucinous and signet ring gastric cancer caused by the relative acellularity and low glucose transporter 1 protein (GLUT1) expression [101]. In the paper by Kuten et al. [48], a comparison between [^68^Ga]Ga-FAPI-04 and [^18^F]FDG PET/CT has been conducted in a group of 13 patients with GAc. For the initial staging evaluation, [^68^Ga]Ga-FAPI-04 PET/CT detected all the primary gastric tumors, while only 50% of cases were [^18^F]FDG positive. The LBR was higher for FAPI even if both tracers detected the same metastatic lymph nodes in two subjects, with an additional lymph node revealed by [^68^Ga]Ga-FAPI-04 PET/CT in a non-glucose-avid cancer. Moreover, in three patients with a poorly differentiated signet ring GAc, FAPI-positive and FDG-negative peritoneal carcinomatosis were reported. Similar results were demonstrated by Gündoğan et al. [49]; in the new diagnosis of GAc, [^68^Ga]Ga-FAPI-04 uptake was observed in the 15 newly diagnosed subjects, while [^18^F]FDG PET/CT did not show any pathological uptake in two patients with mucinous and signet ring cell carcinoma. The sensitivity of metastatic lymph node detection was 100% for [^68^Ga]Ga-FAPI-04 and 71.4% for [^18^F]FDG, while the specificity was 95.2% and 93.7%, respectively. Regarding the peritoneal involvement, [^68^Ga]Ga-FAPI-04 had the same specificity as [^18^F]FDG (100%), and the sensitivity was 100% for [^68^Ga]Ga-FAPI-04 and 40% for [^18^F]FDG [49].

In the study by Lin et al. [50], 45 patients with GAc were recruited for staging and 11 patients for restaging after surgery. Even if the two tracers had comparable abilities in the detection of primary tumor and lymph node metastasis, [^68^Ga]Ga-DOTA-FAPI-04 was more accurate in detecting peritoneal involvement and bone lesions. [^68^Ga]Ga-DOTA-FAPI-04 could better show the primary lesion, the metastatic lymph nodes, and the peritoneal masses thanks to the higher LBR and, for peritoneal lesions probably due to the stromal fibroblastic changes which can occur during peritoneal cancer invasion [102]. The uptake in gastric signet ring cell carcinoma (GSRCC) was higher for [^68^Ga]Ga-DOTA-FAPI-04, as expected, with a higher LBR. Furthermore, FAPI SUVmax and LBR correlated to the primary tumor depth. LBR was higher in groups III–IV, T3–T4, and lower in T1–T2 and groups I–II, demonstrating the capability to predict invasion depth. Moreover, LBR correlated to lymph node invasion since it was lower in the N0 stage than in the N1-3 stage [102].

The superiority of [^68^Ga]Ga-FAPI-04 PET was confirmed by Zhang et al. [51] in 25 patients affected by GAc. The detection rate for primary lesions, pathologic lymph nodes, and distant metastasis and the SUVmax values were higher for [^68^Ga]Ga-FAPI-04 than for [^18^F]FDG; furthermore, [^68^Ga]Ga-FAPI-04 PET/CT revealed new pathological findings in fourteen out of twenty-five cases and modified tumor staging in seven out of twenty-five patients. 

The usefulness of [^68^Ga]Ga-FAPI-04 PET/CT in GSRCC has been systematically investigated in the multicenter study by Chen et al. [52], in which it demonstrated higher sensitivity than [^18^F]FDG in the detection of primary lesions (even though both scans were negative in six subjects with stage IA and median tumor size of 0.5 cm) lymph nodes, bone, and visceral metastasis. Moreover, [^68^Ga]Ga-FAPI-04 PET/CT upstaged three patients, detected more lesions in six patients, displayed [^18^F]FDG-negative local recurrences in five patients and [^18^F]FDG-negative metastasis (located in the peritoneum, lung, lymph node, colon, and pancreas) in seven patients [52].

The bicentric retrospective analysis from Jiang et al. [53] confirmed the superior [^68^Ga]Ga-FAPI-04 detection rate of the primary site (four cases of adenocarcinoma and three cases of signet ring cell carcinoma missed by [^18^F]FDG), whereas the sensitivity in the nodal evaluation was only slightly better than [^18^F]FDG (60% vs. 50%, respectively) [53].

Qin and colleagues were the first to compare the performance of [^68^Ga]DOTA-FAPI-04 PET/MRI to that of [^18^F]FDG PET/CT in GC patients [54], confirming the advantage of [^68^Ga]DOTA-FAPI-04 over [^18^F]FDG for primary tumor detection (100% vs. 71.43%; *p* = 0.034) and in both patient-based and lesion-based evaluation, except for the metastatic lesions in supradiaphragmatic lymph nodes and ovaries. Additionally, multiple sequences of MRI were beneficial for the interpretation of hepatic, uterine, rectal, ovarian, and osseous metastases [54].

Nevertheless, dual-tracer PET/CT seems to be a crucial choice for the complete staging of GC. Indeed, the combination of the two imaging modalities hugely improves the diagnostic sensitivity for detecting distant metastases, according to the work by Miao and colleagues [55]. Even though primary lesions and peritoneal recurrences had higher SUVmax and LBR values for [^68^Ga]Ga-FAPI-04 than [^18^F]FDG, the dual-tracer method seems to be more accurate and permitted to change the treatment approach in nine patients out of 62. Interestingly, [^68^Ga]Ga-FAPI-04 PET/CT outperformed [^18^F]FDG PET/CT in the identification of the lesion borders, especially in poorly cohesive carcinoma. Regarding lymphatic involvement, comparable performance in the detection of regional nodal metastases has been observed between the dual-tracer and single-tracer scans. In the abovementioned study [55], it was also reported that [^18^F]FDG PET/CT detected two lung and two bone [^68^Ga]Ga-FAPI-04-negative metastases. 

#### 3.2.4. Multiple Myeloma

Multiple myeloma (MM) is a neoplastic disease of the bone characterized by uncontrolled clonal proliferation of plasma cells, which causes mainly osteolytic lesions in the skeleton [103]. A whole-body CT is the modality of choice for the initial assessment of MM, and MRI is the gold standard modality for detecting bone marrow involvement [103]. Furthermore, [^18^F]FDG PET/CT provides valuable prognostic data, and it is preferred for evaluating response to treatment [104,105], even though low hexokinase-2 expression in MM may cause false-negative results [106]. For this reason, Elboga and colleagues [56] aimed to compare the performances of [^68^Ga]Ga-FAPI-04 and [^18^F]FDG PET/CT in detecting localizations of the disease in 14 patients with MM but found no significant superiority in [^68^Ga]Ga-FAPI-04 over [^18^F]FDG PET/CT. However, [^68^Ga]Ga-FAPI PET/CT might still be advantageous in low-[^18^F]FDG affinity and inconclusive cases, also considering its low background activity and the absence of non-specific bone marrow uptake [56].

#### 3.2.5. Gastrointestinal Stromal Tumor

Gastrointestinal stromal tumors (GISTs) are a subtype of STS which occur mainly in the stomach (about 60–65%), followed by the small intestine (20–25%), large intestine (4–7%) and esophagus (1%) [107]. CeCT is the standard imaging method for staging and follow-up, except for lesions located in the rectum or in the liver and patients allergic to iodinated contrast agents, where an MRI is preferable [108]. [^18^F]FDG PET/CT is only recommended if CT or MRI are ambiguous, but up to 20% of lesions may show very low or even absent uptake [109,110,111]. Some preliminary results [112,113] considered [^68^Ga]-labeled FAPI PET/CT a promising imaging tool for GISTs; consequently, Wu et al. [57] focused on comparing the detectability of [^18^F]FAPI-42 PET/CT and [^18^F]FDG PET/CT in different recurrent and metastatic sites of GISTs as well as the factors (e.g., tumor size, degree of enhancement, type of gene mutation, targeted treatment) potentially influencing the uptake of the two tracers. In a cohort of 35 patients, a total of 106 lesions were identified, out of which 38/106 (35.8%) were FAPI+/FDG− (26 liver metastases, 10 peritoneal metastases, one gastrointestinal recurrence, and one bone metastasis). The positive detection rate of [^18^F]FAPI-42 PET/CT for recurrent or metastatic GISTs was higher than that of [^18^F]FDG (80.2% vs. 53.8%, *p* < 0.001), especially in liver metastases (87.5% vs. 33.3%, *p* < 0.001) [57]. In addition, they concluded that the uptake of [^18^F]FAPI-42 could reflect the level of FAP expression, and it was independent of tumor size, degree of enhancement, type of gene mutation, and targeted therapy as compared to [^18^F]FDG [57].

#### 3.2.6. Bladder Cancer

The renal clearance and high tracer accumulation in the urinary bladder are limiting factors for the use of [^18^F]FDG PET/CT in primary tumor detection of urothelial carcinoma [114,115]. On the contrary, CAFs promote tumorigenesis in urothelial bladder carcinoma via multiple markers, including alpha smooth muscle actin, CD90/Thy-1, platelet-derived growth factor receptor-alpha and -beta (PDGFR-α/-β) and especially in advanced stages significantly increased FAP-expression [116]. Considering this potential, Novruzov and colleagues [58] sought to evaluate the diagnostic potential of [^68^Ga]FAPI in seven patients with bladder cancer and compare it with that of [^18^F]FDG PET/CT. [^68^Ga]FAPI PET/CT showed to be superior to [^18^F]FDG PET/CT in detecting metastatic lesions in patients with advanced bladder cancer, as it detected an additional 30% (*n* = 9) of lesions missed by [^18^F]FDG and improved tumor delineation [58].

#### 3.2.7. Pancreatic Cancer

Pancreatic ductal adenocarcinoma (PDAC) accounts for more than 85% of cancers, and the incidence has continued to increase over the past decade [80]. The 5-year relative survival rate for PDAC was 3% [117]. Consequently, accurate diagnosis, staging, and early detection of the recurrence or metastasis are essential in the precise management of PDAC. PET/CT with [^18^F]FDG has great advantages in clinical staging, therapeutic evaluation, and detection of tumor recurrence, as it has a sensitivity of 0.80 and a specificity of 0.89 for detecting recurrent pancreatic cancer [118]. By comparison, for CT, the sensitivity and specificity were 0.70 and 0.80, respectively. However, the false-positive uptake of [^18^F]FDG PET/CT in pancreatic cancer was often found in inflammatory diseases, whereas it has a limited role in detecting LN involvement of PDAC [119]. Pancreatic cancer is one of the tumors with abundant CAFs and intermediate radiolabeled FAPI uptake [16].

Liu et al.’s [59] retrospective study enrolled 51 patients suspected to have pancreatic tumors who underwent double-tracer PET/CT. [^68^Ga]Ga-DOTA-FAPI-04 showed a higher sensitivity than [^18^F]FDG for detecting primary tumor (100% vs. 95.0%), metastatic lymph nodes (96.2% vs. 61.5%) and distant metastases (100% vs. 84.0%) (*p* < 0.0001, respectively).

Zhang and colleagues [60] prospectively evaluated 33 patients suspected to have PDAC, of whom thirty-two were confirmed by histopathology, and one had autoimmune pancreatitis confirmed by needle biopsy and glucocorticoid treatment. Within 1 week, each patient underwent both [^68^Ga]Ga-FAPI-04 PET/MR and [^18^F]FDG PET/CT. Thirty pancreatic cancer patients and three pancreatitis patients were enrolled. [^68^Ga]Ga-FAPI-04 PET/MR and [^18^F]FDG PET/CT exhibited equivalent (100%) detection rates for primary tumors. Sixteen pancreatic cancer patients had pancreatic parenchymal uptake, whereas ^18^F-FDG PET images showed parenchymal uptake in only four patients (53.33% vs. 13.33%, *p* < 0.001). [^68^Ga]Ga-FAPI-04 PET detected more positive lymph nodes than [^18^F]FDG PET (42 vs. 30, *p* < 0.001), while [^18^F]FDG PET was able to detect more liver metastases than [^68^Ga]Ga-FAPI-04 (181 vs. 104, *p* < 0.001). In addition, multisequence MR imaging helped explain ten pancreatic cancers that could not be definitively revealed due to [^68^Ga]Ga-FAPI-04 inflammatory uptake and identified more liver metastases than [^18^F]FDG (256 vs. 181, *p* < 0.001).

### 3.3. Tumors with High [^18^F]FDG Uptake

#### 3.3.1. Lung Cancer

Despite the well-established role of [^18^F]FDG PET/CT in the staging process of lung oncologic disease, there are some limitations to the clinical accuracy of such tracer, making [^18^F]FDG PET/CT a highly sensitive (94% to 96%) but less specific (78% to 86%) technique at staging and restaging [120]. One of the most influential factors is that some slow-growing or less active cancer histotypes (such as mucinous adenocarcinoma or neuroendocrine tumors) are less FDG-avid, leading to false-negative results [121]. On the contrary, some factors could lead to false-positive results, such as infectious or inflammatory processes (i.e., sarcoidosis, tuberculosis, pneumonia, silicosis) and surgical or post-radiation changes [122].

Due to the presence of contradictory findings in the literature, it is still unclear whether [^18^F]FDG and FAPI exhibit meaningful differences in the detection of primary tumors [61,123,124], but in consideration of both a higher SUVmax and LBR in metastases, some studies reported that the diagnostic accuracy of [^68^Ga]FAPI PET/CT is almost 100%, compared with a value up to 89.6% for [^18^F]FDG PET/CT, particularly in the detection of brain, lymph nodes, bone, and pleura metastases [61,62,63,64] thanks to a higher LBR [16,125]. 

As a limitation, it should be highlighted that FAPI-based tracers may not be advantageous in cases of pulmonary fibrosis because lung cancer and interstitial lung disorders both exhibit increased uptake of [^68^Ga]- or [^18^F]FAPI; this confounding factor should be taken into consideration when interstitial diseases should be involved [126].

#### 3.3.2. Head and Neck Cancer

Head and neck cancer (HNC) represents one of the top common tumor types, with an incidence of greater than half a million cases diagnosed annually [80]. Over 90% of all HNCs are squamous cell carcinomas (HNSCC) with poor prognosis at advanced tumor stages. Nasopharyngeal carcinoma (NPC) is an epithelial carcinoma arising from the nasopharyngeal mucosal lining; it is an aggressive cancer with extensive local spread and a high incidence of cervical nodal and distant metastases. Combining functional and anatomical imaging, [^18^F]FDG PET/CT has been widely and successfully applied in diagnosis, staging, restaging, and recurrence detection for these types of tumors. Recently, several studies compared the diagnostics performance of [^68^Ga]Ga-FAPI PET/CT with [^18^F]FDG PET/CT to explore the clinical impact (TNM staging) in HNC. Linz et al. [65] presented a pilot study to investigate the feasibility of staging newly diagnosed, treatment-naive oral squamous cell carcinoma (OSCC) patients using [^68^Ga]Ga-FAPI-04 PET/CT and to compare its diagnostic performance with [^18^F]FDG PET/CT. Authors demonstrated identical primary tumor detection by the two tracers; while regarding metastases, they showed a slightly lower sensitivity as well as a marginally higher specificity for cervical lymph node involvement of [^68^Ga]Ga-FAPI-04 in comparison to [^18^F]FDG PET/CT. Promteangtrong et al. [66] performed the first head-to-head comparison of diagnostic performance and semiquantitative parameters between [^68^Ga]Ga-FAPI-46 and [^18^F]FDG PET/CT in HNSCC patients, demonstrating that there were no differences in the assessment of both TNM staging and recurrent detection between the two tracers with 83.3% and 96.4% concordance, respectively. Additionally, their findings showed that there were no significant differences in semiquantitative parameters, except for the FAPI expression tumor volume (FTV) of the primary tumor, namely the equivalent value of the [^18^F]FDG-metabolic tumor volume, namely the MTV, assessed by [^68^Ga]Ga-FAPI-46, which was significantly higher than the MTV of the primary tumor [66]. Similarly, the study by Chen et al. [67] demonstrated no differences between [^68^Ga]Ga-FAPI-46 and [^18^F]FDG PET/CT to identify the primary lesion but showed a better specificity and accuracy of [^68^Ga]Ga-FAPI-46 for the detection of OSCC neck lymph node metastases than [^18^F]FDG PET/CT. These data were confirmed by Jiang et al. [68], who showed that [^68^Ga]Ga-FAPI-04 PET/CT has a comparable diagnostic performance with [^18^F]FDG for detecting primary tumors and local recurrence in patients with HNSCC and that [^68^Ga]Ga-FAPI-04 PET/CT presents higher specificity and accuracy for preoperative N staging compared to [^18^F]FDG PET/CT. Several prospective studies that compared the diagnostic performance of [^68^Ga]Ga-FAPI PET/CT with [^18^F]FDG PET/CT in NPC detection demonstrated that [^68^Ga]Ga-FAPI PET/CT had a positive impact on the clinical stage of NPC [69,70,71]. Since [^68^Ga]Ga-FAPI PET/CT had better LBR than [^18^F]FDG PET/CT and less false-positive uptake in inflammatory and reactive proliferative lymph nodes, it improved the capability to recognize primary cancer and lymph node metastases, mainly for the assessment of the skull base and intracranial invasion. Nevertheless, when it comes to the detection of distant metastases, [^68^Ga]Ga-FAPI PET/CT does not show any advantage over [^18^F]FDG PET/CT [69,70,71]. Wegen et al. [72] demonstrated that [^68^Ga]Ga-FAPI-46 PET/CT in HNC had promising features and might be at least equivalent to [^18^F]FDG PET/CT in terms of accuracy and useful not only for staging purposes but also (and especially) for radiotherapeutic treatment planning. The hybrid imaging modality [^18^F]FDG PET/MRI is well known to be superior to [^18^F]FDG PET/CT in case of evaluation of HNC [127,128,129]. Qin and colleagues [71] aimed to compare [^18^F]FDG and [^68^Ga]FAPI PET/MRI in a population of 15 NPC, showing that, compared to [^18^F]FDG, [^68^Ga]FAPI PET/MRI improved the delineation of skull-base and intracranial invasion due to its low brain background uptake; however, its value regarding lymph node evaluation needs further study, as significantly more [^18^F]FDG-avid lymph nodes were detected than those of [^68^Ga]FAPI (100 vs. 48) [71].

Eight patients with suspicion of a malignant tumor in Waldeyer’s tonsillar ring or a CUP syndrome were examined with both [^68^Ga]Ga-FAPI-04 and [^18^F]FDG PET/CT [73]. The first resulted in better visual detection of the malignant primary in CUP, as compared to [^18^F]FDG imaging. However, the detection rate of lymph node metastases was inferior, presumably due to low FAP expression in small metastases. Nevertheless, by offering a detection method for primary tumors with the potential of lower false-positive rates and thus avoiding biopsies, patients with CUP syndrome may benefit from [^68^Ga]FAPI PET/CT imaging.

Finally, [^68^Ga]Ga-FAPI PET/CT could improve the detection rate of the primary tumor in HNC of unknown primary origin patients with negative [^18^F]FDG PET/CT findings; furthermore, for evaluating metastatic lesions, [^68^Ga]Ga-FAPI PET/CT showed a similar performance to [^18^F]FDG PET/CT [74].

#### 3.3.3. Esophageal Cancer

With approximately 600,000 new diagnoses and 540,000 deaths in 2020, esophageal cancer (EC) is the sixth leading cause of cancer-related death worldwide [80]. Squamous cell carcinoma (SCC) is the predominant histological type of esophageal cancer in East Asia, East and Southern Africa, and Southern Europe, while adenocarcinoma is most common in Northern and Western Europe, Oceania, and North America [80]. [^18^F]FDG PET/CT is a currently used imaging modality for staging, restaging and treatment response assessment, although its main limitation is the low to moderate sensitivity for lymph node staging and delineation between viable tumor and regional esophagitis [130]. Case studies reported the application of [^68^Ga]Ga-DOTA-FAPI-04 PET/CT in the detection of primary tumors and metastatic lymph nodes in EC [75,131,132]. In addition, Liu and colleagues [76] retrospectively evaluated 35 patients with newly diagnosed EC or surgical resected EC (34 SCC and one adenocarcinoma) who underwent both [^68^Ga]Ga-DOTA-FAPI-04 and [^18^F]FDG PET/CT. [^68^Ga]Ga-DOTA-FAPI-04 PET/CT showed a higher detection sensitivity than [^18^F]FDG PET/CT for primary tumors (100% vs. 96.0%), lymph nodes (95.0% vs. 75.0%, *p* < 0.001), bone and visceral metastases (pleural less than 1 cm, 100% vs. 72%, *p* = 0.008) [76]. 

#### 3.3.4. Colorectal Cancer

Colorectal cancer (CRC) was reported as the fifth most common cause of cancer-related deaths in the United States in 2022 [133]. The overall 5-year survival of patients with CRC largely depends on the stage at presentation, varying from 80–90% in the early stages to 13% in the advanced stage [134,135]. Optimal imaging for CRC is crucial for accurate initial staging and the selection of primary therapy as well as during follow-up examinations for the accurate and timely detection of local recurrence and/or metastasis. Moreover, in this setting, [^18^F]FDG PET/CT has several limitations, including low specificity, inability to detect small volume lesions and lack of isotope uptake in mucinous and signet ring cell carcinomas [136]. Again, FAPI-based tracers have demonstrated the ability to overcome the limitations of [^18^F]FDG. In particular, Lin et al. [77] evaluated 61 patients with metastatic CRC lesions who underwent sequential evaluation through PET/CT with [^68^Ga]Ga-FAPI-04 and [^18^F]FDG. They found that the average LBR and SUVmax in signet-ring/mucinous carcinomas primary CRC lesions or in peritoneal and liver metastases were higher on [^68^Ga]Ga-FAPI-04 PET images than in [^18^F]FDG PET images and, more importantly, [^68^Ga]Ga-FAPI-04 PET/CT led to upstaging and downstaging in ten (16.4%) and five participants (8.2%), respectively compared to [^18^F]FDG, and the treatment options were changed in thirteen participants (21.3%) [77]. Similar results were obtained in the study from Kömek and colleagues [78] in 39 patients with histopathologically confirmed primary or relapsed CRC, proving that [^68^Ga]Ga-DOTA-FAPI-04 PET/CT achieved higher sensitivity and specificity compared to [^18^F]FDG in primary lesions detection, and, especially, the lymph nodes (sensitivity of 90% vs. 80% and specificity of 100% vs. 81.8%, respectively) and peritoneal metastases (sensitivity 100% vs. 55%, respectively), suggesting that it can be employed in the assessment of primary tumor and metastases in patients with CRC in routine clinical practice. Furthermore, in all lesions, the LBR was higher in [^68^Ga]Ga-DOTA-FAPI PET/CT, whereas the SUVmax of primary lesions was higher with [^18^F]FDG (*p* < 0.001) but lower in lymph nodes [78].

#### 3.3.5. IgG4-Related Disease

Immunoglobulin (Ig)G4-related disease (IgG4-RD) is characterized by lymphoplasmacytic infiltration enriched in IgG4-positive plasma cells and variable degrees of fibrosis with a characteristic storiform pattern [137]; due to these hallmarks, Luo and colleagues [79] speculate that it was possible for IgG4-RD to be imaged with [^68^Ga]Ga-FAPI, as already reported by some case reports [79,138], and purposed to compare this new imaging modality to the reference standard, i.e., [^18^F]FDG PET/CT, in a population of 26 patients observing a higher detection rate or higher uptake than [^18^F]FDG in IgG4-RD in most involved organs, especially in the pancreas, bile duct/liver, lacrimal gland, and salivary gland [139]. However, IgG4-related lymphadenopathy was not FAPI–avid, a finding that may be attributed to the fact that IgG4-related lymphadenopathy usually lacks the characteristic storiform fibrosis [139].

## 4. Discussion

FAPI PET tracers are characterized by some benefits over [^18^F]FDG, such as higher LBR, fast renal clearance and tracer kinetics, no dependency on blood glucose and resting. 

Based on the available information, certain conclusions can be drawn regarding the comparison of FAPI and [^18^F]FDG, despite the fact that the exact role of FAPI in some FDG-avid malignancies remains unclear. For instance, FAPI could have a complementary role in detecting the presence of brain metastases in lung cancer, but further studies are needed to confirm this data.

FAPI is more sensitive than [^18^F]FDG for the detection of primary tumors, lymph nodes, and metastatic lesions in gastrointestinal cancer, including esophageal cancer and colon–rectal cancer, which may often display a high-to-moderate diffuse uptake that may reduce lesion detectability. 

Moreover, in female malignancies, including breast cancer and ovarian cancer, radiolabeled FAPI demonstrated a larger accumulation than [^18^F]FDG in both primary tumors and metastatic lymph nodes. Namely, many comparison studies showed that for the same SUVmax, LBR is significantly higher, resulting in a better detection of lesions. 

Radiolabeled FAPI may also play a role in the early diagnosis of head and neck cancer, particularly when the original tumor site is unknown.

Considering some types of cancers in which the role of [^18^F]FDG still remains controversial, FAPI PET has been reported to have diagnostic advantages in detecting both intrahepatic and extrahepatic HCC and non-HCC cancers. This is due to its demonstrated higher sensitivity and accuracy than [^18^F]FDG, along with higher SUVmax and LBR in liver disease. 

FAPI could also be employed for several other clinical indications, such as sarcoma or urothelial cancer, which are now frequently understaged by traditional molecular imaging due to limited sensitivity in the case of low-grade tumors or sites with physiologic elevate uptake. 

Moreover, FAPI has been shown to be more accurate than [^18^F]FDG for the detection of peritoneal carcinomatosis. In fact, [^18^F]FDG exhibits a natural biodistribution in the intestinal loops that can consideringly diminish the detection of peritoneal carcinomatosis, particularly in the omental cake pattern. The advantage of radiolabeled FAPI over [^18^F]FDG in the detection of peritoneal carcinomatosis is a fundamental characteristic which can increase tumor resectability and may affect the prognosis of the patient. 

Moreover, the complementary role of the two radiopharmaceuticals should be taken into consideration. For instance, FAPI would probably have a complementary role in detecting the presence of brain metastases in lung cancer, but further information studies are needed to confirm this data. In addition, the dual-tracer PET/CT seems to be a crucial tool for the complete staging of gastric cancer, improving patients’ management and guiding treatment strategy.

However, the available research has revealed certain limitations and different issues to be verified regarding radiolabeled FAPI. For example, the potential clinical application of FAPI PET in detecting lesions and guiding radioligand therapy of thyroid cancer remains controversial due to a relatively high false-negative rate reported in some studies and opposite controversial reports in other articles as a promising tool for detecting metastatic lesions.

One significant drawback, less evident than [^18^F]FDG, is the high rate of false-positives when inflammatory processes are present. Similarly, there are no significant data for the evaluation of response to treatment, especially for the verification of fibrotic changes occurring in some responding lesions vs. the persistence of residual disease. Moreover, several studies have not used histopathology as the standard of reference to determine the advantages of FAPI-based tracers over [^18^F]FDG. Finally, because of the small number of patients enrolled in most of the studies, often retrospectively recruited, additional research is necessary to assess the appropriate alternative role of FAPI in various solid neoplasms.

In conclusion, radiolabeled FAPI is a promising novel radiopharmaceutical agent with distinct advantages over [^18^F]FDG, and we may speculate that in the near future, FAPI-based tracers could be complementary to or even a substitute for radiolabeled glucose in several clinical scenarios, as already hypothesized at the very beginning of their appearance in clinical studies [140,141].

One of the most notable aspects of FAPI is its potential application as a theranostic agent. At present, the most promising clinical data of FAPI theranostics has been reported in advanced sarcoma, breast, thyroid, and pancreatic cancer [142,143,144,145]. Therefore, it is critical to collect more evidence and design appropriate clinical trials to gain a better understanding of FAPI’s exact role and applications in a theranostic setting.

## Figures and Tables

**Figure 1 life-13-01821-f001:**
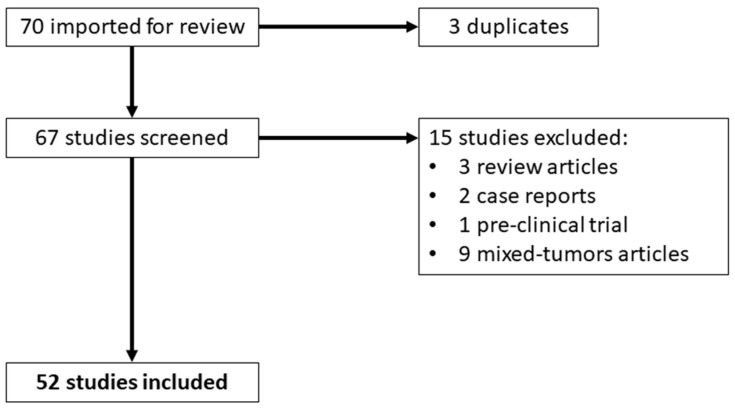
PRISMA flow-chart.

**Figure 2 life-13-01821-f002:**
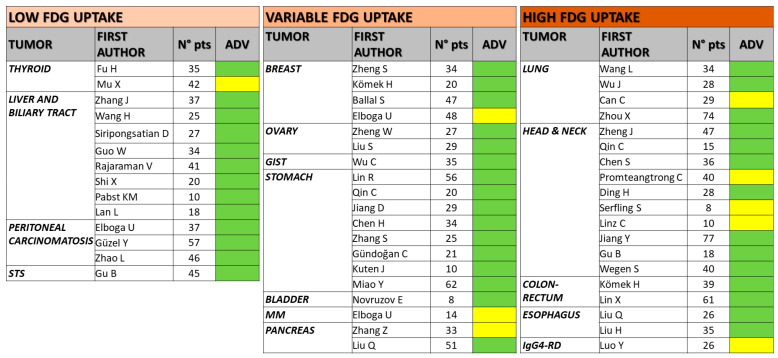
Schematic representation of the studies and their advantage in terms of detection rate and/or accuracy. pts: patients; ADV: advantage of FAPI-based tracers over FDG (green: strong advantage; yellow: only moderate advantage or comparable results); STS: soft tissue sarcoma; MM: multiple myeloma; IgG4-RD: IgG4-related [28,29,30,31,32,33,34,35,36,37,38,39,40,41,42,43,44,45,46,47,48,49,50,51,52,53,54,55,56,57,58,59,60,61,62,63,64,65,66,67,68,69,70,71,72,73,74,75,76,77,78,79].

**Figure 3 life-13-01821-f003:**
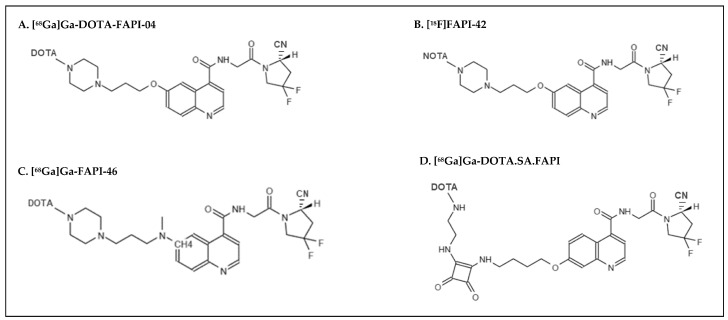
Structures of the FAPI-based radiotracers used in the studies.

**Figure 4 life-13-01821-f004:**
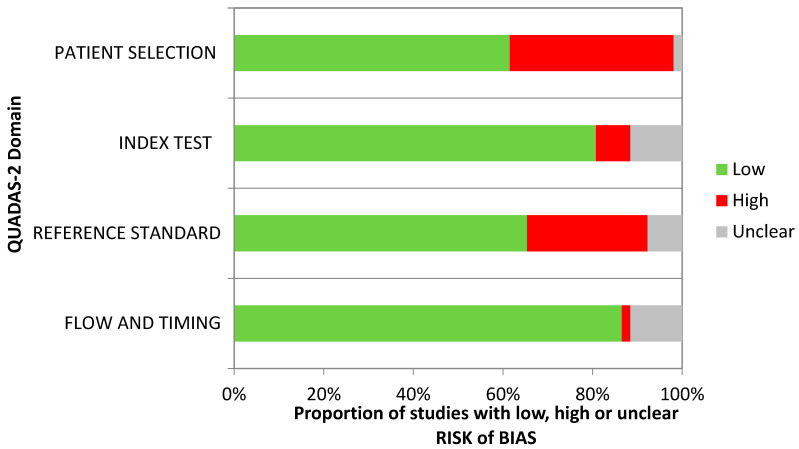
QUADAS-2 domain.

**Table 1 life-13-01821-t001:** Main characteristics of the included studies. P: prospective; R: retrospective; n.a.: not available. Y: yes; N: no; D: doubtful [28,29,30,31,32,33,34,35,36,37,38,39,40,41,42,43,44,45,46,47,48,49,50,51,52,53,54,55,56,57,58,59,60,61,62,63,64,65,66,67,68,69,70,71,72,73,74,75,76,77,78,79].

First Author	Journal	Year	Country	FAPI-Based Tracer Used	Study Design	Primary Tumor Site	Clinical Indication	Reference Standard	Number of Patients (M:F)	Median (Range) or Mean (±SD) Age	Metastases (Y/N)	Advantage over [^18^F]FDG
Fu, H.	Radiology	2022	China	[^68^Ga]Ga-DOTA-FAPI-04	P	Thyroid	Suspect of distant metastasis	Histopathology OR clinical/imaging follow-up	35 (18:17)	44 (28–58)	Y	Y
Mu, X.	EJNMMI	2023	China	[^18^F]FAPI-42	P	Thyroid	Detection of recurrence	Imaging	42 (16:26)	46 (12–75)	Y	D
Zhang, J.	EJNMMI	2023	China	[^18^F]FAPI	P	Liver	Diagnostic accuracy	Histopathology OR clinical/imaging follow-up	37 (34:3)	57 (48–67)	Y	Y
Wang, H.	Frontiers in Oncology	2021	China	[^68^Ga]Ga-DOTA-FAPI-04	R	Liver	Diagnostic accuracy	n.a.	25 (24:1)	59.40 ± 6.90	Y	Y
Siripongsatian, D.	Molecular Imaging and Biology	2022	Thailand	[^68^Ga]Ga-FAPI-46	R	Liver	Diagnostic accuracy	MRI images	27 (21:6)	68 (60–74)	Y	Y
Guo, W.	EJNMMI	2021	China	[^68^Ga]Ga-DOTA-FAPI-04	R	Liver	Diagnostic accuracy	Histopathology OR clinical/imaging follow-up	34 (25:9)	60.6 (33–75)	Y	Y
Rajaraman, V.	Clinical Nuclear Medicine	2023	India	[^68^Ga]Ga-DOTA-FAPI-04	P	Liver and biliary tract	Diagnostic accuracy	n.a.	41 (28:13)	n.a.	Y	Y
Shi, X.	EJNMMI	2021	China	[^68^Ga]Ga-DOTA-FAPI-04	P	Liver	Diagnostic accuracy	Histopathology OR clinical/imaging follow-up	20 (18:2)	58.0 ± 9.4	Y	Y
Pabst, K.M.	JNM	2023	Germany	[^68^Ga]Ga-FAPI-46	P	Biliary tract	Staging	Histopathology OR clinical/imaging follow-up	10 (6:4)	55.5 (40–79)	Y	Y
Lan, L.	Radiology	2022	China	[^68^Ga]Ga-DOTA-FAPI-04	P	Biliary tract	Diagnostic accuracy	Histopathology OR clinical/imaging follow-up	18 (6:12)	61 ± 10	Y	Y
Elboga, U.	Molecular Imaging and Biology	2022	Turkey	[^68^Ga]Ga-DOTA-FAPI-04	R	Peritoneal carcinomatosis	Diagnostic accuracy	Histopathology OR clinical/imaging follow-up	37 (23:14)	62.8 ± 12.7	Y	Y
Güzel, Y.	Hellenic Journal of Nuclear Medicine	2023	Turkey	[^68^Ga]Ga-DOTA-FAPI-04	R	Peritoneal carcinomatosis	Diagnostic accuracy	CT images	57 (25:32)	54 (22–86)	Y	Y
Zhao, L.	EJNMMI	2021	China	[^68^Ga]Ga-DOTA-FAPI-04	R	Peritoneal carcinomatosis	Diagnostic accuracy	Histopathology OR clinical/imaging follow-up	46 (14:32)	57 (32–80)	Y	Y
Zheng, S.	Clinical Nuclear Medicine	2023	China	[^68^Ga]Ga-DOTA-FAPI-04	P	Breast	Initial staging	Histopathology	34 F	51.5 (36–72)	Y	Y
Kömek, H.	Annals of Nuclear Medicine	2021	Turkey	[^68^Ga]Ga-DOTA-FAPI-04	P	Breast	Staging OR detection of recurrence	n.a.	20 F	44 (32–65)	Y	Y
Ballal, S.	Pharmaceuticals	2023	India	[^68^Ga]Ga-DOTA.SA.FAPI	R	Breast	Diagnostic accuracy	Histopathology OR clinical/imaging follow-up	47 F	44.8 ± 9.9	Y	Y
Elboga, U.	Annals of Nuclear Medicine	2021	Turkey	[^68^Ga]Ga-DOTA-FAPI-04	R	Breast	Diagnostic accuracy	n.a.	48 F	53.3 ± 11.7	Y	Y
Zheng, W.	Nuclear Medicine Communications	2022	China	[^68^Ga]Ga-DOTA-FAPI-04	R	Ovary	Staging OR detection of recurrence	Histopathology OR clinical/imaging follow-up	27 F	56 ± 12	Y	Y
Liu, S.	EJNMMI	2023	China	[^68^Ga]Ga-DOTA-FAPI-04	P	Ovary	Detection of recurrence	n.a.	29 F	56.90 ± 8.38	Y	Y
Wegen, S.	Clinical Nuclear Medicine	2023	Germany	[^68^Ga]Ga-FAPI-46	R	Cervical	Initial staging	Histopathology	7 F	57 (34–67)	Y	Y
Wu, C.	EJNMMI	2022	China	[^18^F]FAPI-42	R	GIST	Detection of recurrence or distant metastases	Histopathology OR clinical/imaging follow-up	35 (21:14)	54 (32–76)	Y	Y
Lin, R.	EJNMMI	2022	China	[^68^Ga]Ga-DOTA-FAPI-04	P	Stomach	Diagnostic accuracy	Histopathology OR clinical/imaging follow-up	56 (40:16)	63.8 ± 14.9	Y	Y
Qin, C.	JNM	2022	China	[^68^Ga]Ga-DOTA-FAPI-04	P	Stomach	Diagnostic accuracy	n.a.	20 (9:11)	56 (29–70)	Y	Y
Jiang, D.	EJNMMI	2022	China	[^68^Ga]Ga-DOTA-FAPI-04	R	Stomach	Diagnostic accuracy	Histopathology	29 (20:9)	67 (25–86)	Y	Y
Chen, H.	European Radiology	2023	China	[^68^Ga]Ga-DOTA-FAPI-04	R	Stomach	Diagnostic accuracy	Histopathology OR clinical/imaging follow-up	34 (16–18)	51 (25–85)	Y	Y
Zhang, S.	Frontiers in Oncology	2022	China	[^68^Ga]Ga-DOTA-FAPI-04	R	Stomach	Diagnostic accuracy	Histopathology OR clinical/imaging follow-up	25 (12:13)	56 ± 12	Y	Y
Gündoğan, C.	Nuclear Medicine Communications	2021	Turkey	[^68^Ga]Ga-DOTA-FAPI-04	P	Stomach	Diagnostic accuracy	n.a.	21 (12:9)	61 (40–81)	Y	Y
Kuten, J.	EJNMMI	2022	Israel	[^68^Ga]Ga-DOTA-FAPI-04	P	Stomach	Detection rate	Histopathology OR clinical/imaging follow-up	10 (n.a.)	71 (42–87)	Y	Y
Miao, Y.	European Radiology	2022	China	[^68^Ga]Ga-DOTA-FAPI-04	P	Stomach	Diagnostic accuracy	Histopathology OR clinical/imaging follow-up	62 (44:18)	64 (24–75)	Y	Y
Wang, L.	Radiology	2022	China	[^68^Ga]Ga-FAPI	P	Lung	Diagnostic accuracy	MRI images (brain)	34 (20:14)	64 (46–80)	Y	Y
Wu, J.	Frontiers in Oncology	2022	China	[^68^Ga]Ga-FAPI	P	Lung	Staging	Histopathology OR clinical/imaging follow-up	28 (13:15)	60.5 (34–78)	Y	Y
Can, C.	Nuclear Medicine Communications	2022	Turkey	[^68^Ga]Ga-DOTA-FAPI-04	R	Lung	Diagnostic accuracy	n.a.	29 (27:2)	71 (46–84)	Y	D
Zhou, X.	EJNMMI	2022	China	[^68^Ga]Ga-DOTA-FAPI-04	P	Lung	Diagnostic accuracy	Histopathology OR clinical/imaging follow-up	74 (39:35)	63 ± 8	Y	Y
Zheng, J.	Molecular Imaging and Biology	2022	China	[^68^Ga]Ga-DOTA-FAPI-04	P	Nasopharyngeal	Diagnostic accuracy	CT and MRI images	47 (32:15)	52.3 ± 13.8	Y	Y
Qin, C.	EJNMMI	2021	China	[^68^Ga]Ga-DOTA-FAPI-04	P	Nasopharyngeal	Diagnostic accuracy	MRI images	15 (8:7)	51.2 ± 9.4	Y	Y
Chen, S.	European Radiology	2022	China	[^68^Ga]Ga-DOTA-FAPI-04	P	Oral cavity	Preoperative staging	Histopathology	36 (29:7)	62.5 (34–87)	Y	Y
Promteangtrong, C.	JNM	2022	Thailand	[^68^Ga]Ga-FAPI-46	P	Head and neck	Diagnostic accuracy	Histopathology OR clinical/imaging follow-up	40 (27:13)	57.5 (32–86)	Y	D
Wegen, S.	Molecular Imaging and Biology	2022	Germany	[^68^Ga]Ga-FAPI-46	R	Head and neck	Radiotherapy planning	n.a.	15 (12:3)	66 (37–82)	Y	D
Ding, H.	Frontiers in Oncology	2022	China	[^68^Ga]Ga-DOTA-FAPI-04	P	Nasopharyngeal	Initial staging	Histopathology OR imaging follow-up	28 (23:5)	53 ± 11	Y	Y
Serfling, S.	EJNMMI	2021	Germany	[^68^Ga]Ga-DOTA-FAPI-04	R	Waldeyer’s tonsillar ring	Suspect of disease	n.a.	8 (n.a.)	62 (58–72)	Y	D
Linz, C.	EJNMMI	2021	Germany	[^68^Ga]Ga-DOTA-FAPI-04	P	Oral cavity	Initial staging	Histopathology	10 (8:2)	62 ± 9	Y	D
Jiang, Y.	EJNMMI	2023	China	[^68^Ga]Ga-DOTA-FAPI-04	P	Head and neck	Diagnostic accuracy	Histopathology OR clinical/imaging follow-up	77 (61:16)	58 (20, 89)	Y	Y
Gu, B.	JNM	2022	China	[^68^Ga]Ga-DOTA-FAPI-04	P	Head and neck	Detection FDG- findings	Histopathology	18 (16:2)	55 (24–72)	Y	Y
Liu, H.	Frontiers in Oncology	2022	China	[^68^Ga]Ga-DOTA-FAPI-04	R	Esophagus	Diagnostic accuracy	Histopathology OR clinical/imaging follow-up	35 (32:3)	63.5 (44–83)	Y	Y
Kömek, H.	EJNMMI	2021	Turkey	[^68^Ga]Ga-DOTA-FAPI-04	P	Colon or rectum	Diagnostic accuracy	Histopathology	39 (22:17)	61 (29–83)	Y	Y
Lin, X.	Frontiers in Oncology	2023	China	[^68^Ga]Ga-DOTA-FAPI-04	P	Colon or rectum	Diagnostic accuracy	Histopathology OR imaging follow-up	61 (42:19)	62 (32–81)	Y	Y
Liu, Q.	European Radiology	2023	China	[^68^Ga]Ga-DOTA-FAPI-04	R	Pancreas	Diagnostic accuracy and prognostic stratification	Histopathology OR clinical/imaging follow-up	26 (19:7)	62.4 ± 11.9	Y	Y
Zhang, Z.	EJNMMI	2022	China	[^68^Ga]Ga-DOTA-FAPI-04	P	Pancreas	Suspect of disease	Histopathology	33 (19:14)	63 (46–81)	Y	D
Novruzov, E.	Molecular Imaging and Biology	2022	Germany	[^68^Ga]Ga-DOTA-FAPI-04	R	Bladder	Diagnostic accuracy	Histopathology (only bladder)	8 M	66 (57–78)	Y	Y
Gu, B.	EJNMMI	2022	China	[^68^Ga]Ga-DOTA-FAPI-04	P	Soft tissue	Detection of recurrence	Histopathology OR clinical/imaging follow-up	45 (24:21)	46 (18–71)	Y	Y
Luo, Y.	JNM	2021	China	[^68^Ga]Ga-DOTA-FAPI-04	P	IgG4-related disease	Diagnostic accuracy	FDG PET/CT	26 (20:6)	51.5 ± 12.9	Y	D
Elboga, U.	Tomography	2022	Turkey	[^68^Ga]Ga-DOTA-FAPI-04	R	MM	Detection rate	Histopathology OR clinical/imaging follow-up	14 (7:7)	58 (39–81)	Y	D

## Data Availability

Not applicable.

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
