# Peer review of "Head-to-Head Comparison of FDG and Radiolabeled FAPI PET: A Systematic Review of the Literature"

_life, 2023, doi:10.3390/life13091821_

Round 1

Reviewer 1 Report

Summary: The manuscript entitled “Head-to-head Comparison of FDG and Radiolabeled FAPI PET: A Systematic Review of The Literature” screened 52 researches about the comparison between FAPI tracer and 18F-FDG for PET imaging in several cancers, which compared the two tracers in detail based on cancer types and summarized current study progress on FAPI imaging in patients. This study is suitable for publication in Life.

Comments:

1. Could the authors include the structures of current FAPI tracer such as FAPI-04, -42, -46, -74, etc. so that the readers can easily clarify which ones were used in current studies?

2. Correct some typos such as “In example” at line 713, page 11.

None

Author Response

We thank the reviewer for the valuable comments. Here below a point-by-point response:

1. Could the authors include the structures of current FAPI tracer such as FAPI-04, -42, -46, -74, etc. so that the readers can easily clarify which ones were used in current studies?

We added the following sentence in the Results section:

Of these 52 articles, 41 papers were focused on [68Ga]Ga-DOTA-FAPI-04 (Figure 3A), two papers on [18F]FAPI-42 (Figure 3B), five papers on [68Ga]Ga-FAPI-46 (Figure 3C), one on [68Ga]Ga-DOTA.SA.FAPI (Figure 3D) and in three studies only the radioisotope used (68Ga or 18F) was declare.

Subsequently, we added the images of the four structures.

2. Correct some typos such as “In example” at line 713, page 11.

We carefully revised the whole text in order to correct typos. Please see the attachment.

Reviewer 2 Report

In recent years, the tumour microenvironment has gained growing attention in the context of complementary diagnostic and therapeutic strategies in oncology. Fibroblast activation protein inhibitor (FAPI) used for PET imaging is a strategy that targets the cell population of the cancer-associated fibroblasts. The protein itself - a membrane-bound type 2 serine protease. It is a member of the dipeptidyl peptidase 4 family and represents a specific surface marker.

Positron emission tomography is a powerful diagnostic imaging tool in oncology mainly due to the application of 18F-fluorodeoxyglucose (FDG). FDG is a glucose analogue that is accumulated in most cancer cells to a higher degree than in normal cells of the body. However, FDG-PET is not entirely specific for the detection of tumours as all inflammatory lesions avidly accumulate the tracer. Therefore, other, more sensitive and specific radioligands are necessary for cancer imaging. I guess, discovery of FAPI ligands has the highest impact on the field of PET imaging after the introduction of the Hamacher method in the production of FDG. Therefore, the head to head comparison of the new ligand and the workhorse of the PET imaging is really important. I can realize two definitive advantage of FAPI's over the FDG. Improved specificity, due to FDG has high physiological uptake in normal tissues (e.g. brain), low uptake in some tumour types (e.g. well-differentiated neuroendocrine tumours), and lack of specificity to differentiate inflamed tissues from malignant lesion - but in some cases the role of FAPI ligands are also controversial. The second is the potential role of FAPI's in the theragnostic concept.

I can agree with the authors that there are promising clinical data with the new ligands, but more evidence needs to evaluate the real value of the tracers. I guess, AI will make the real ground-breaking, due to a higher number of designs will eliminate of the subjective parts of the current studies. 

Independently of these remarks I believe that this article is very useful and well organised, and really useful for those who wants a deep review about this topic. I can congratulate for the authors for their profound summary. I can find only a really few misspellings, but it is a great work.

Author Response

We warmly thank the reviewer for the compliments. 

Furthermore, we revised the text in order to correct all the typos. Please see the attachment.
